# On the Time Scales of Optical Variability of AGN and the Shape of Their Optical Emission Line Profiles

**Edi Bon [1],*** , **Paola Marziani [2], Predrag Jovanović [1] and Nataša Bon [1,2]**

[1]  Astronomical Observatory, Volgina 7, 11060 Belgrade, Serbia; pjovanovic@aob.rs (P.J.); nbon@aob.rs (N.B.)
[2]  National Institute for Astrophysics (INAF), Astronomical Observatory of Padova, IT 35122 Padova, Italy; paola.marziani@inaf.it
*  Correspondence: ebon@aob.rs

**Abstract:** The mechanism of the optical variability of active galactic nuclei (AGN) is still very puzzling. It is now widely accepted that the optical variability of AGN is stochastic, producing red noise-like light curves. In case they were to be periodic or quasi-periodic, one should expect that the time scales of optical AGN variability should relate to orbiting time scales of regions inside the accretion disks with temperatures mainly emitting the light in this wavelength range. Knowing the reverberation scales and masses of AGN, expected orbiting time scales are in the order of decades. Unfortunately, most of monitored AGN light curves are not long enough to investigate such time scales of periodicity. Here we investigate the AGN optical variability time scales and their possible connections with the broad emission line shapes.

**Keywords:** AGN; black holes; gravitational waves; binary black holes; quasars

## 1. Introduction

Active galactic nuclei (AGN) are very strong and variable emitters [1]. It is widely believed that the AGN patterns correspond to red noise-like curves [2,3], as a result of unpredicted processes of fluctuations of their accretion disks (AD). In some cases, these variations appear periodic, with periodogram peak significance jumping above the red noise levels (see for example, [4]). Unfortunately, most AGN monitoring campaign time intervals are not sufficiently long for detecting periodicity in optical domain of spectra, since expected orbiting time scales in AGN ADs that would affect the optical part of the spectrum are of the order of years and decades. Luckily, significant periodicities are detected in several AGN which are extensively monitored for sufficiently long time, such as OJ287 [4,5], NGC 4151 [6–8], NGC 5548 [9,10], Ark 120 [11], 3c273 [12,13].

It is expected that in galaxy mergers, their cores should eventually end up close to each other, and get gravitationally bounded in the close-orbiting system of supermassive black holes (SMBHs). Such configurations are called supermassive binary black hole (SMBBH) systems, with period of several to tens of years resulting from orbiting timescales of the system.

Some numerical simulations of SMBBHs show expected periodic behaviors of their light curves [14–17].

### 1.1. Variability Time Scales and Amplitudes

Main variability time scales of AGN are light crossing (order of days to months), orbiting (years to decades), and sound speed (order of hundreds of years) [18].

The amplitude of variation is different in each object. The variability of radio loud and radio quite types seems to be more or less similar, within the one sigma of difference [19]. The variability is most significant in case of some "changing look"(CL) objects, such as NGC 5548 [9], NGC 4151 [7],

and some other examples see, (e.g., [20,21]). Some of them appear to be periodic, such as for example NGC 5548, NGC 4151, and a few more periodic CL candidates; see [7,9,22].

It is not yet clear what triggers such variability that can lead to changing type. Investigating the periodic CL case of NGC 5548 in the context of Eigenvector 1, showing that besides that this object is mainly Pop B1, it also varies in long term observations lead to conclusion that the main driver of the variability could be connected to the accretion rate changes, and the obscuration effects [23]. The evidence of intrinsic obscuration within the broad line region itself, as also found for a recent very short epoch in monitoring of this object [24]. How these two mechanisms combined could produce periodical variability is still not clear.

### 1.2. Periodic Variability of AGN

Mechanisms proposed to explain the periodic emission variability of AGN are: jet/outflow precession, disk precession, disk warping, orbiting of spiral arms, flares, and other kinds of instabilities orbiting within the accretion disk, tidal disruption event, the existence of a SMBBH system in their cores see, [9,25–32]...

AGN are very hard to prove to be periodic [3]. Therefore, standard methods such as Fourier and Lomb-Scargle [33,34] may show peaks of high looking significance but the derived p-value may not be valid [3,9,35]. There are a few historic AGN light curves spanning over 100 years (monitored first as variable stars, before recognized to be distant galaxies) found to show significant periodicity of order of years to decades see (e.g., [4–6,13]).

### 1.3. Supermassive Black Hole Binaries

A probable explanation of periodic variability is the possibility of SMBBH, since such scenario should be able to produce most significant variations of accretion rates, as well as gas ejections and obscuration effects.

The orbiting variability time scales in AGN are of an order of several decades. Unfortunately, it is hard to find cases of AGN light curves with several repeating patterns [4,5,9,10,29,30,36], needed for the clear detection of periodicity, above the red noise level, since the length of current AGN monitoring campaigns are of the order of orbiting time scales. Therefore, as an indicator of orbiting effects we could be tracing the broad emission line shifts, and if the periodicity in radial velocity curves is the same, it could indicate that the mechanism which drives both curves could be linked to the orbiting within the broad line region [7,9–11,31,37]. Radial velocity curves are harder to obtain due to even shorter records of spectral observations, and therefore only a few candidates are detected, such as NGC 4151 with a 15.9 year periodicity [6,7], NGC 5548 with a ≈15 year periodicity [9,10], and Ark 120 with a ≈20 year periodicity [11]. It is interesting that current results of all these cases show that the radial velocity curves of the red side of broad emission lines show larger amplitude shift with more significant indication of periodicity, as in case of NGC 3516 [38], where the same effect is observed in red wing of Fe K$\alpha$ line. However, this is expected, since that the gravitational effects are more significant on the red side of the line due to gravitational redshift effect [39], and it could be seen in magneto hydrodynamic simulations of eccentric SMBBH systems [27].

Periodic light curve candidates also include a blazar OJ 287, with 11.5-year period, as the most famous SMBBH candidate [4,5], the quasar PG 1302-102 (6 year period, [29]), the blazar PG 1553+113 (2 year period, [40]), a 13- and 21-year periodicity found in 3C 273 [12,13].

The light variations may not be the only direct effect of the moving SMBHs. The effects of SMBBH system orbital motion should be also affecting the jet bending, which could be used to model the SMBBH system properties with jet observations (see, for example [41–44]) Unfortunately, quasi-periodicities in the jet emission can be induced by intrinsic oscillatory disk instabilities that can mimic periodical behavior.

Most significant effects to light curves are expected from orbiting SMBHs. Simulations of such emission produced in SMBBH systems is very complex, and therefore mainly tested for simplified

configurations, such as comparable mass SMBHS, and nearly circular orbits [45,46]. Only a few simulations are currently available for the eccentric high-mass ratio systems [27,28,47,48].

*1.4. Broad Emission Line Shape Connection with the Variability Time Scales*

The connection of AGN variability with broad emission line shape changes was suggested in several papers see, e.g., [23,38,49–51]. The orbiting gas is expected to be in some form of flattened distribution [52–56] that could be surrounded with isotropic gas component [57–60].

Orbiting gas properties are closely related to the mass of the central supermassive body, gas distribution, and inclination of the system. Therefore, the shape of broad emission lines could give us some constraint of the gas configuration and the central BH mass.

Recently, one such model was proposed [23], where they investigated the connection between the variability time scales of active galactic nuclei (AGN) optical light curves with the shapes of their broad emission line profiles. Knowing that the perturbing region orbital period is related to the SMBH mass and to the radial distance of the perturbing region [38,51], they propose that the variability time scale of the optical continuum light curves could be connected with the perturbing region located at the part of a disk seen as the inner and outer radius for the optical broad Balmer lines. Using the accretion disk model [52] as a tool to measure the disk size and parameters (like the inclination angle), and connecting it with the variability time scales of the AGN light curves, they were able to obtain similar masses of the central BHs, as from reverberation campaigns. From virial mass and the orbiting calculated with this model it is possible to calculate the inclination of the emitting gas orbiting plane. In [23] the obtained inclination angle from their method agreed with the inclination angle obtained by the disk model fitted in the broad emission line profiles. This could indicate that the inner and outer radii of an accretion disk might be indeed connected with the AGN variability time scales.

In case the variability time scales relate to the orbiting time scales of SMBBH systems, then identification of such systems could be very important for gravitational wave (GW) observations [61]. In the epochs of last orbits, the gas could be squeezed producing super-Eddington outflows [61].

## 2. Method

Knowing that the light curves of a continuum at 5100 Å and broad H$\beta$ emission line are highly correlated [62] may indicate the same origin of their variability. Therefore, the source that drives the variability could leave a trace in the shapes of their broad lines. Analysis of variation time scales may give us valuable information about why they vary the way they do, while the line profiles could provide us with the information about the kinematic parameters of the variability drive (such us the radii where the source of variation is located).

To investigate the variability patterns, we used periodicity analysis of optical continuum light curves, with hope to find periodic or quasi-periodic variability patterns, with time scales corresponding to the orbital time scales within the region where the optical light could be originating inside the AD. If we could detect signatures in the broad emission line profiles, which could be produced by the effects of the same phenomena that drives the variability of the same time scale for that peculiar periodic pattern, then we could be able to determine dynamical properties of such AGN. In case we could identify more than one significant period in their light curves that could be linked to the radius of amplified ring emission in the broad emission line shape, then for each ring-radius pair we should expect to obtain the same mass (or at least very close value) of the central BH mass using a Kepler's laws.

To test these assumptions, as a first step we model synthetic line emission of an orbiting gas in the flat, disk-like gas distribution, assuming that photo-ionization processes produce the emission line from that region, that we could approximate with the accretion disk emission model [39,49,52–54,56,63,64].

By matching the disk model with the line profile, we determine the inclination, and inner and outer radii. In the shapes of BELs sometimes could be found small bump-like features that cannot be modeled

with such disk models, assuming emissivity parameter that would correspond to photo-ionization (and assuming a negligible mass of a perturbed region in the ring compared to the central SMBH).

Simulated profiles of AD emission usually have characteristic two peaks in the core of the line, while the wings are broadened due to relativistic effects. The two peaks are usually blended by the isotropic emission component, located away from the AD, which is present in majority of AGN spectra [57–60,65,66]. Only in a very small number (less than 1%) of objects the two peaks are clearly recognized see e.g., [54,67].

*Model of AD with Amplified Thin Ring Emission*

The AD model is an idealization of emission with assumption of homogeneous AD that may not be the case, and therefore is not sufficient to describe all features in observed profiles, like for e.g., small bumps in the wings of the line profiles that are often present. To describe these additional features (like bumps in the broad H$\beta$ emission line profiles) we assumed additional amplified emission component located in thin rings inside the AD. We model the thin ring emission assuming it is located somewhere in the interval from 100–4000 R$_g$ ( which we vary in the model with a step of 100 R$_g$), assuming the ring width of 100 R$_g$. We add the contribution of each ring profile to the AD model profile as an additive combination of rings profiles, with its intensity multiplied by a scaling factor (see Figures 1–3).

Assuming that the time scale of perturbed disk (cooling time, shock wave progression, or anything that produced additional emission from that ring) is significantly longer than the orbital time scale, then we could approximate that the quasi-periodic variations found in optical light curves correspond to the orbiting of some features inside of the AD at radii that we can associate with amplified thin rings that we can locate by matching the emission lines and synthetic modeled profiles. Their radii are measured in units of gravitational radii Rg, since from the AD model we cannot obtain the information about the central mass. By connecting each period to thin ring-radius with the assumption that the shorter period corresponds to closer ring, and the outer ring corresponds to a longer period, we could be able to calculate the mass of the central SMBH using the Kepler's third law for a circular orbit:

$$P = 2\pi\sqrt{\frac{r^3}{GM}} = \frac{2\pi GM\xi^{3/2}}{c^3},\tag{1}$$

where $r = \xi(r_g)$ is the ring-radius in gravitational radii and $P$ is the circular orbital period of the orbiting region at such radii see, [23,50,51].

We decided to use the relativistic ray-tracing AD model[1] see [39,64,68], to build a new, more complex model, of an AD assuming the amplified thin ring emission. We assumed that the origin of the variability patterns, detected in quasi-periods, could be traced back in the broad line profiles (seen as amplified thin rings emission inside of AD at different radii). Making a connection of periods and radii, using the Equation (1), could allow us to open a new window into dynamical properties of AGN.

Our model assumes the emission of an AD and ring models as a linear combination of contributions to the line profiles with the same parameters as in the parent AD, which are preserved in ring models. To construct our model, we simulated the grid of models of the $H\beta$ line profiles using the code which includes both special relativistic and general relativistic effects on radiation from the accretion disk around SMBH see e.g., [68]. This AD model is based on the ray-tracing method in the Kerr metric [63,64], for different values of inner and outer radii and inclinations of rings in AD. The emissivity index was fixed as q = −2, assuming the emissivity law to be ∼$r^q$, as expected for the case of photo-ionization. The model is then constructed using a previous match of the AD profile to the emission line, as a starting point. Then the scaled contributions of the thin ring profiles are added to the AD profile until bumpy features in observed spectra are described with the synthetic spectrum.

---

[1]　We tested several different AD models [39,52,56,63,64,68] for a line fit, and found that obtained inclinations were practically the same regardless of the model used.

Beside the fact that the shape of the line is fitted more realistic then with a simple AD model, we are also obtaining a valuable information about the radii in the disk, where the emission is amplified.

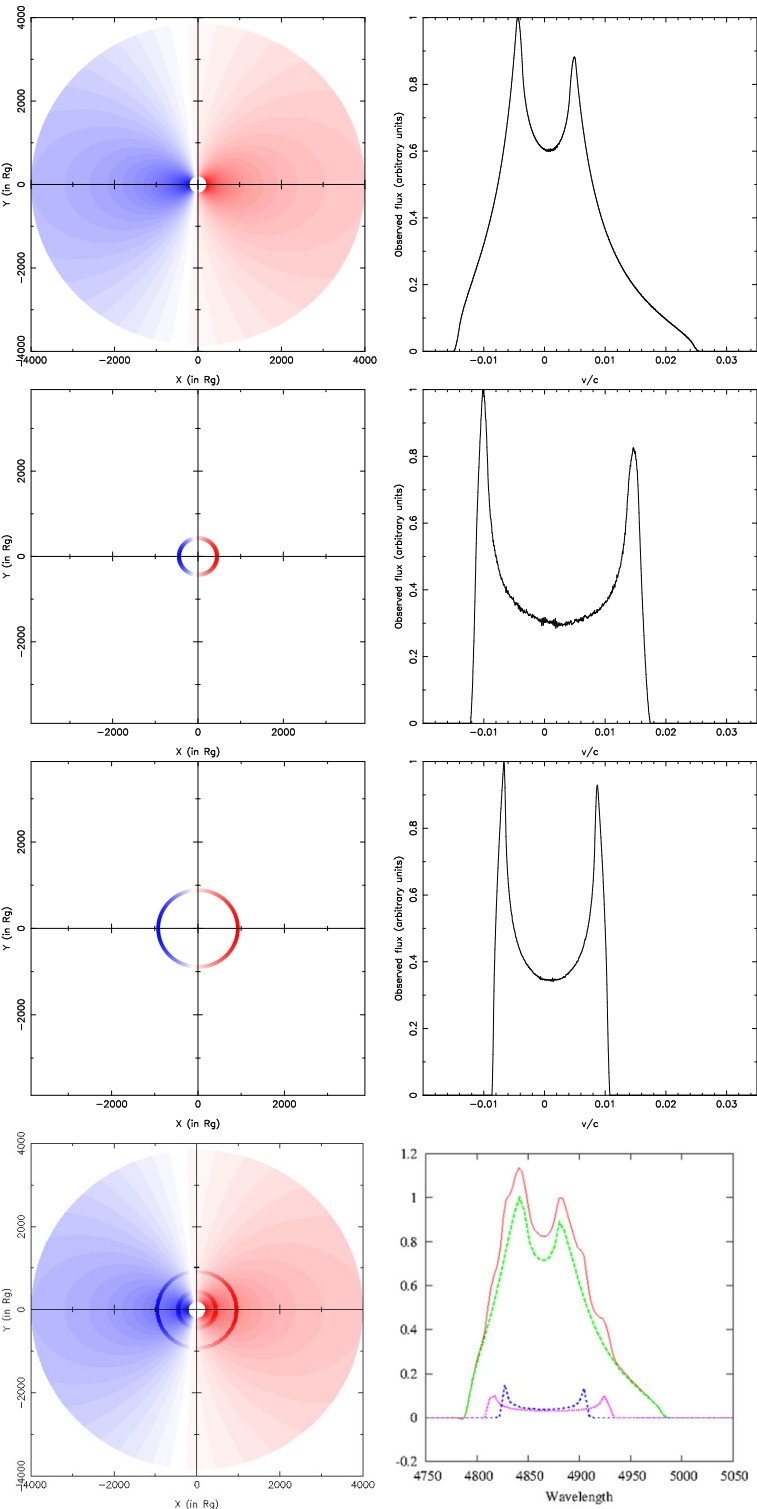

**Figure 1.** Construction of a synthetic profile as a linear combination of components: AD ($R_{inn}$ = 100 $R_g$, $R_{out}$ = 8000 $R_g$, top panel right) and two additional synthetic ring profile models ($R_{inn1}$ = 500 $R_g$, $R_{out1}$ = 600 $R_g$, $R_{inn}$ = 1000 $R_g$, $R_{out2}$ = 1100 $R_g$ mid right panels respectively). The resulting emission line profile is presented as right bottom panel as sum of contribution of AD and two ring profiles with intensity scaled by a factor of 0.1. Representations of disk and ring models are presented on the right panels.

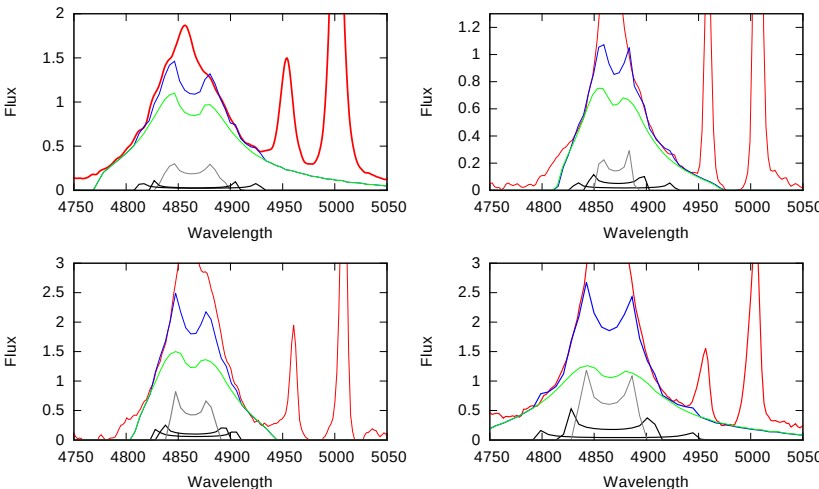

**Figure 2.** A few examples of AD + ring model matching the broad Hβ emission line profiles.

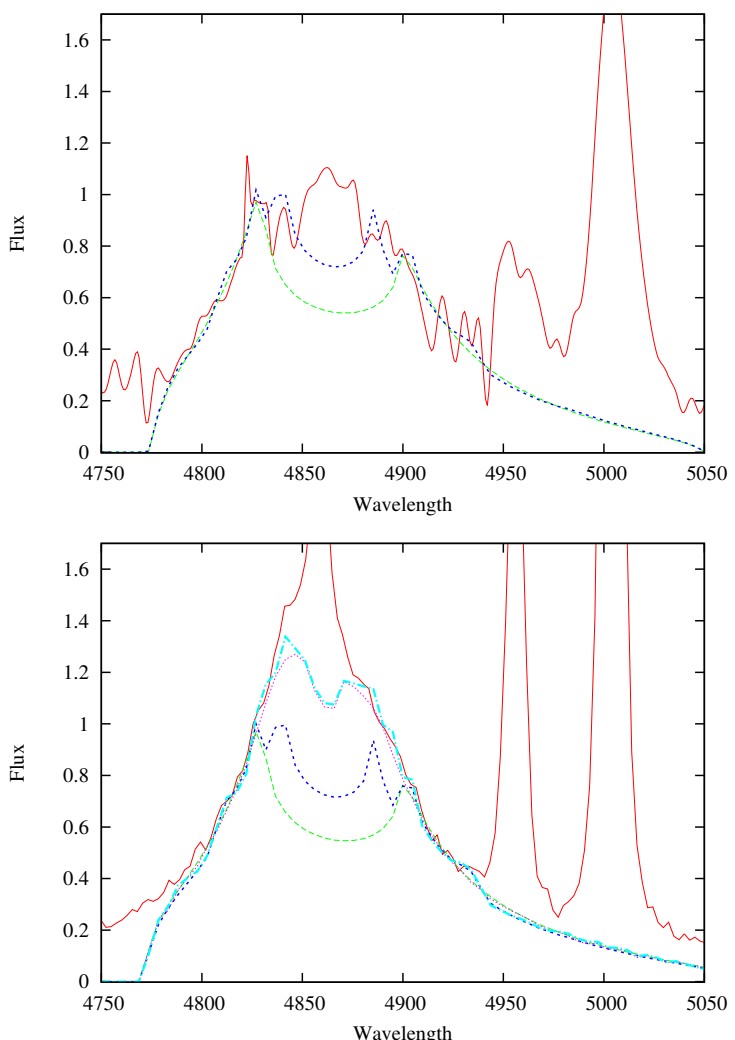

**Figure 3.** A few examples of AD + ring model matching the monitoring broad Hβ emission line profiles of a single object in low (**top** panel) and high (**bottom** panel) state. Both profiles are matched with models with same parameters, except for the outer radius, which is 1500 $R_g$ for the low state profile (**top**) and 8000 $R_g$ for the higher state line profile (**bottom** panel). The model indicates that the difference between these two states is in ionization of outer BLR, with radii above 1500 $R_g$.

One example of how the profiles are constructed is presented in Figure 1. As could be seen, by adding small contributions of ring profiles to the AD line shape, we make a model for the same inclination. In this peculiar case, the inclination angle is assumed to be 16 degrees. The AD was calculated with $Rinn = 200\ R_g$ and $Rout = 4000\ R_g$. The velocity distribution over the surface of AD in ray-tracing model is presented on the left, while the line shape binned from it is presented on the right panel. For the rings we assumed the same inclination, emissivity law, and the disk shift. We simulated two ring emission profiles, assuming $Rinn_1 = 500\ R_g$ to $Rout_1 = 600\ R_g$ and $Rinn_2 = 1000\ R_g$ to $Rout_2 = 1100\ R_g$, and added them to the AD profile assuming the multiplying factor of 0.1. The resulting profile could be seen in the bottom panel of Figure 1.

For the periodicity analysis we used standard Lomb-Scargle method [33,34]. Obtained periods are fitted afterwards with linear combination of sine functions, as an additional test of results.

## 3. Results

To test this hypothesis, we selected a sample presented in [62], where all spectra and light curves were publicly available. Also, for the periodicity analysis, beside light curves from [62], we used additional light curves from Catalina Real-Time Transient Survey see [29,30,36]. We used only single epoch spectra instead of averaged, to avoid smoothing by averaging. In some cases, we used spectra of the same objects, from [69] paper, due to better S/N.

Using the radii measured from line profile matching to the model, we measure the ring radii that we hoped to be connected to the variability quasi-periods, as a measurement of variability time scales. Assuming that the orbital time scale is the only match to time scale of variability patterns seen in these light curves, we combine measured radii and periods, and derived mass assuming circular Keplerian orbit of this brighter region positioned in the ring of the AD. We test the model by calculating masses of the central BH, with expectation that the obtained results for masses, for each pair of period and radius, should be equal, or at least of similar or close values.

We measured ring radii in units of gravitational radii ($R_g$) from the line profiles, and detected two or more significant periods, with a threshold at 400 days period to avoid effects of Earth's orbital period of 1 year. Unfortunately, expected orbital period in optical part of the AD for typical AGN of $M \sim 10^8\ M_\odot$ is about one year. Therefore, any amplification in the AD profile under $300\ R_g$ was also avoided, since it would correspond to such time scales, or even shorter periods.

Here we do not consider any details about what produces the hot ring region in the AD, and we are fully aware that the measured periods are significant above the standard, white noise levels, but may not appear significant compared to the red noise AR curves [2,3]. We were mainly interested in measuring time scales of orbital periods assuming that the variability patterns [19] in the light curves could be induced by the orbital time scales.

Examples of model in the line profile matched to single epoch spectral profiles are presented in Figure 2. PG 0052 in low and high state is presented as an example of an object spectroscopically monitored. The low- and high-state profiles are matched with the same parameters, except for the contribution of the outer part of the disk with upper limit of $1500\ R_g$. This radius in this case is the one we connected with the longer period, while the shorter period is paired with radius of $1000\ R_g$ (of the amplified ring making bumps at the core of the profile next to the narrow OIII [4959] line, see Figure 3). These bumps are even better recognized in higher S/N spectrum of the same object presented at the top right panel of the Figure 2, where one can see the observed spectrum in red, model of an AD+rings (blue), just AD model (green), and each of the additional ring contributions (black) that are paired with periods. The outer part rings are presented with gray line[2]. The model indicates that the difference

---

[2]　This outer contribution would correspond to much longer period that we could not be able to detect with such short monitoring intervals of several years, in case it could be connected to any periodicity. These distant radii are probably excited due to reverberation processes.

between these two states of this monitoring is in ionization of outer BLR, with radii above 1500 R$_g$. Measuring this radius, and pairing it with characteristic periodicity can allow us to calculate the mass, assuming that we know the inclination from the fit of the model, as well.

Therefore, assuming that ring radii are connected to quasi-periods obtained from light curves, we calculate central BH mass, for each pair of period-radius. Results of calculated masses are presented on the plot against FWHM in the Figure 4. Calculated masses from each period–radius pair result with similar value of mass (see Figure 4), that is, what we should expect if the hypothesis of the model is correct. Therefore, obtaining close values of mass estimates is justifying the model hypothesis.

Also, getting practically identical values for the mass is very important for the error estimates, showing that this method therefore may be more precise than the reverberation mapping technique.

We note that the obtained masses with our method are mainly in a good agreement with previous results obtained from reverberation mapping results [62]. This could be seen in Figure 5, where we presented the ratio between masses obtained with our method and with the reverberation mapping method given in [62]. The ratio with reverberation of masses obtained from mean profiles are given in orange, while the ratio with masse from RMS profiles are given in blue. These differences of masses obtained with our method and reverberation mapping method (see Figure 5) could be due to the fact that in reverberation mapping technique, the inclinations of each objects are not taken into account when calculating masses. The assumption that FWHM is not corrected for the inclination, or not weighted properly for the contributions of virial and isotropic component, may influence the "f" factor in the virial formula that was assumed to be the same for all objects. Also, the luminosity measurements were assumed to be connected with the BLR radius, which is another approximation in reverberation mapping techniques, and which may add additional systematic effects as well. We note that the largest discrepancy is seen on parts of the plot that is most affected by the inclination effect, see Figure 5 for FWHM under 2000 km/s, and above 8000 km/s, since the "f" factor in virial formula was assumed to correspond to the averaged inclination angle in all sources.

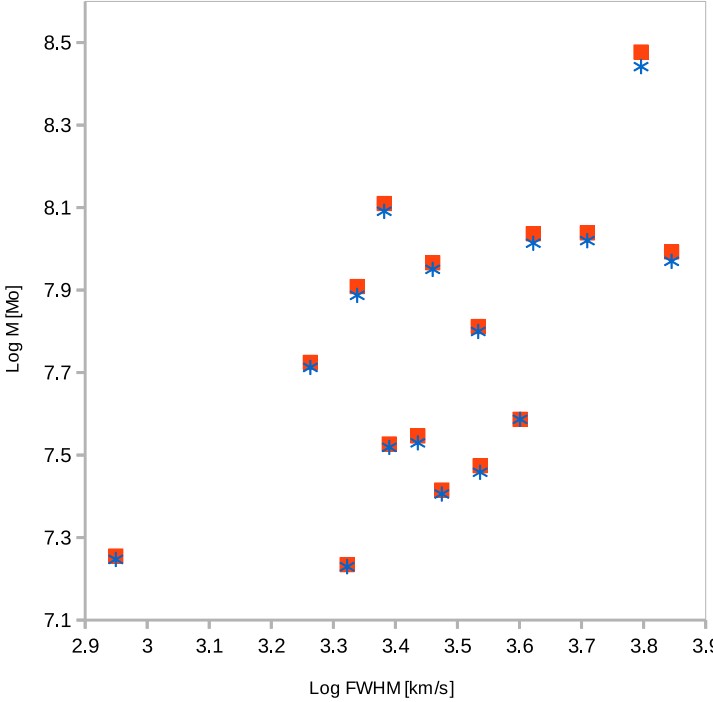

**Figure 4.** The Full Width at Half Maximum of H$\beta$ line versus mass of central SMBH, calculated for each radius-period pair (orange squares and blue asterisks).

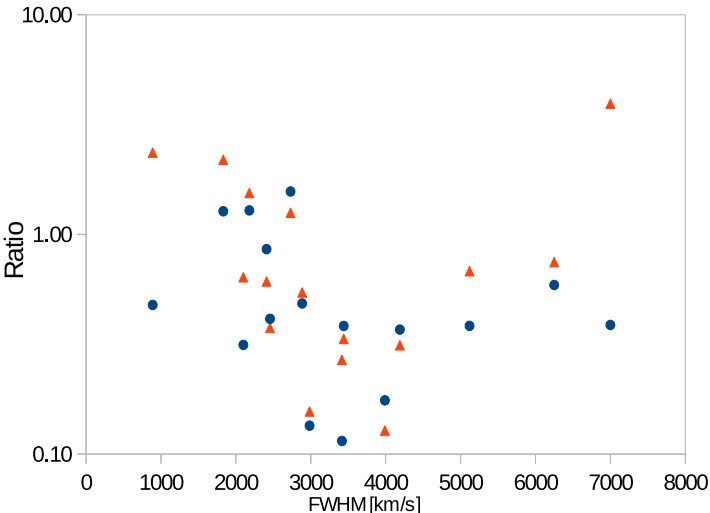

**Figure 5.** The comparison of SMBH masses obtained in this paper with previous reverberation mapping results from Kaspi et al. [62]. Blue full circles represent the ratio of masses from this work and the masses from the reverberation results obtained using FWHM of averaged line profiles, while the orange triangles correspond to ratio with reverberation masses from the RMS profiles.

## 4. Possible Interpretations

Possible interpretations of periodicities are discussed in many works [7,9–11,30,31,35,36,50,51,54,70]. Assuming circular orbits in the disk as we did here, we suggest that possible source of periodicity should be in the AD, amplifying an emission contribution at that radius. We are aware that at such radii, the standard models of thermal emission of AD [71,72] shows that the temperature of the disk is relatively low, under 1000 K, which is not sufficient enough for the photo-ionization mechanism to produce optical broad emission lines, or to significantly contribute to the optical continuum flux, without additional emission mechanism, like shocks [50,54,70], hot spot [49,51], secondary orbiting object on a circular orbit around the central SMBH with additional accretion mechanism that is sufficient to produce significant contribution to the continuum and the line emission see e.g., ([73–75], where their MHD simulations show a fast forming of intermediate mass BH's in AD on a circular orbit). It is expected that in such cases the voids or gaps would be formed see, for e.g., [73,76] in the AD (similar to the planet formation in the stellar disks), with pilling up of matter at the outer border of the gap ring see, ([73,76], and the references therein) that may be the region that could be associated with the amplified ring emission that we modeled here.

## 5. Conclusions

We simulated an AD emission profiles with amplified thin ring regions and matched them with the observed profiles of the broad Hβ emission lines, of PG quasars from the sample selected in [62]. With periodicity analysis, we find significant periodicity from the available optical light curves. We pair each ring profile with the periodicity. Using Kepler's third law, we then calculate central SMBH masses.

Our results show:

(1) The model of an AD with amplified thin ring emission could describe the observed Hβ emission line profiles.

(2) Masses calculated from each pair of period and thin ring, result with very similar values (with the discrepancy of less than 20%), justifying the model initial hypothesis and indicating that the features fitted in the line profiles are not some random noisy patterns, but features that could be connected to the real physical properties of the BLR emission.

(3) The obtained masses are with the same order of magnitude as obtained from the reverberation mapping method, with the discrepancy that could be addressed to the lack of use of inclination angle correction in the reverberation mapping analysis.

(4) Even though, we do not go into a detailed interpretation of the origin of periodic variability, this method could be used on single epoch spectra of central mass BH determination, combined with long term photometric monitoring.

We plan to extend the sample (Bon et al. 2019, in preparation), selecting more AGN with long term monitoring data.

**Author Contributions:** Conceptualization, E.B. and P.M.; Methodology, E.B., P.J., P.M. and N.B.; Software, P.J., N.B., P.M. and E.B.; Validation, E.B., P.M., N.B. and P.J.; Formal Analysis, E.B., N.B. and P.J.; Investigation, E.B., P.M. and N.B.; Data Curation, E.B., P.M., N.B. and P.J.; Writing—Original Draft Preparation, E.B. and P.M.; Writing—Review and Editing, E.B. and P.M.; Visualization, E.B. and P.J.; Supervision, E.B. and P.M.; Project Administration, E.B.

**Funding:** This research was funded by the Ministry of Education, Science and Technological Development of the Republic of Serbia, through the project 176003 "Gravitation and the large-scale structure of the Universe" and 176001 "Astrophysical spectroscopy of extragalactic objects". We also wish to acknowledge the COST Action CA16104 "GWverse", supported by COST (European Cooperation in Science and Technology).

**Acknowledgments:** This research is supported by the Ministry of Education, Science and Technological Development of the Republic of Serbia, through the project 176003 "Gravitation and the large-scale structure of the Universe" and 176001 "Astrophysical spectroscopy of extragalactic objects". We also wish to acknowledge the COST Action CA16104 "GWverse", supported by COST (European Cooperation in Science and Technology).

**Conflicts of Interest:** The authors declare no conflict of interest.

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
