# Peer review of "On the Time Scales of Optical Variability of AGN and the Shape of Their Optical Emission Line Profiles"

_atoms, doi:10.3390/atoms7010026_

Round 1

Reviewer 1 Report

I think the paper is too short for readers to understand on such a complicated topic, more details of published results and views should be clearly introduced and classified in the review. The abstract is also poor, as I think this is a review work (I didn't find new results presented in the paper), not a research work, the abstract should summarise recent progress and new results in the field.

In my view this is a good topic to review but curent review is incomplete for the timescales of optical variability of AGN and the shape of their optical line profiles, about two pages in content is too short (e.g. section 1 and 2 are too short or incomplete). I'd suggest to consider all aspects of optical variability of AGN, and related timescales and line profile shapes, etc, for different types of AGN, not only change-look ones and AGNs with binary BHs, e.g. optical variability resulted from Doppler-boosting jets in AGN for blazars, accretion rate changes and instability of accretion disk/corona, and magnetic reconnection in jet/disk, etc. Secondly, the connection between the emission line profiles and variability timescales was not clearly introduced (section 4) in current paper, this part is very interesting and could be rewritten clearly/logically with adding more details. There are some places where should be improved in English.

Author Response

In order to investigate the broad emission line profiles with variability time scales, we developed a new model which connect the variability time scales and accretion disk radii. Using the observations we test the model and give some preliminary results. We added 9 pages with 5 figures to explain the model and results, with conclusions and future work. We focused here only on measuring the variability time scales that showed quasi periodic behaviour, so therefore we did not consider any other mechanism apart from connection to the orbital motion, and changes in the accretion disk structure. 

Reviewer 2 Report

In this work, the authors presented an investigate the AGN optical variability time scales and their possible connections with the broad emission line shapes. The results are interesting/important and I would like to recommend publications of this manuscript in Atoms Sp. Issue SPIG 2018 but with few requests.

Some of the required modifications that appear to be needed:

The authors claim that the inner and outer radii of an Accretion disk is connected with the AGN variability time scales. This statement should be supported by analysis and discussion. It would be a good idea to present it graphically. Also, the authors should make an proper section Conclusion.

Minor corrections:

-  Authors should give full affiliation e.g.

1 Astronomical Observatory, Belgrade, Serbia=> 1 Astronomical Observatory, Volgina 7, 11060 Belgrade, Serbia

Typos:

in Abstract: stil => still;  puzzuling =>  puzzling; tepertures =>  temperatures; wavelnght => wavelength; Unfortunatelly => Unfortunately

Line 33. Page 2: insert space  “… other examples[see, … “    =>    “… other examples [see, … “

Same goes for line 90, p3.

Line 99 p3. :  “Using the the accretion disk model[74] …  “  =>  “Using the accretion disk model [74] …    “

Line 46:  “… cores [see, 9,14,35,42,43,47,56])...”  =>   “… cores [see, 9,14,35,42,43,47,56]).” 

L.85, p 3. : “… [see, for e. g. 29,49,68,70,76].” =>   “… [see e. g. 29,49,68,70,76].”

L 105, p.3:   “…wave (GW) observations [30]”  => “…wave (GW) observations [30].”

L 109: equaly => equally

L 113, p 3.:  write proper Conflicts of Interest or remove it.

-In References:  All references should be written in the same way, according to the journal style (insert article title, … etc.). For Journal Name authors should use Abbreviated Journal Name.

The article is certainly interesting for Atoms Sp. Issue SPIG 2018 but I would recommend moderate revisions.

Respectfully,

Author Response

In order to investigate the broad emission line profiles with variability time scales, we developed a new model which connect the variability time scales and accretion disk radii. Using the observations we test the model and give some preliminary results. We added 9 pages with 5 figures to explain the model and results, with conclusions and future work.

We corrected the article according to the referee suggestions, except for the references, since the article is already very long, so the adding of the title names of each reference would extend the article significantly (there are 109 references).

Round 2

Reviewer 1 Report

The paper has been significantly revised according to the referee's advices, now it has good shape and content, I would suggest to accept it, but there are still some english speling errors and figure's format: as indicated below:

line 1: still, not stil; line 2: remove 'opinion'; line 6: supposed to be;

line 16: spans are; line 18: are of the order; line 46: remove )...; line 73: remove 2nd ); line 76: remove (; line80: are; line 84: was; line 85: remove 'for' before e.g.; line 86: remove 69; line 96: [?]: I suggest to remove all question mark in the paper; line 102: accretion; line 116: accretion disk (AD); line 119: than, not then; line 136: remove ); line 148: than; line 154: corresponds; line 162: the equation 1; line 221: model; line 268: Masses; line 292: ..7, ?;

In the Fig.3 caption: The model indicates.

The letter size in Fig.5 is much larger than other figures, it is better to use similar sizes in all figures. Finally, please check references are correct and have good format in the references list, and I strongly suggest to remove question mark in the citation of reference in the text.

The end.

Author Response

We thank the referee for the comments. We corrected everything as suggested by the referee.

Reviewer 2 Report

From the scientific point of view, the authors have greatly improved the manuscript with additional material, discussion, figures…

It is a pity that there are a lot of the technical shortcomings and problems. Figs. are inside section References. Still there are lot of typos.

References are randomly listed. After Ref [17] is [32] and after [32] is [57]..?  Some Refs. are missing in the text [?]. 

Also, all references should be written in the same way, according to the journal style (insert article title, … etc.). For Journal Name authors should use Abbreviated Journal Name.  In general, it could be useful to check all references.

Therefore, I can not recommend acceptance in the current form.

Respectfully,

Author Response

We thank the referee for the comments. We tried to correct the format of all references and the order of them, as they appear in the text.